Darwin review

evolution, genetics

genome duplication, polyploidy, autopolyploid, physiology, evolution, adaptation

**Author for correspondence:**
Kirsten Bomblies
e-mail: kirsten.bomblies@biol.ethz.ch

# When everything changes at once: finding a new normal after genome duplication

Kirsten Bomblies

Institute of Molecular Plant Biology, Department of Biology, ETH Zürich, Zürich, Switzerland

KB, 0000-0002-2434-3863

Whole-genome duplication (WGD), which leads to polyploidy, is implicated in adaptation and speciation. But what are the immediate effects of WGD and how do newly polyploid lineages adapt to them? With many studies of new and evolved polyploids now available, along with studies of genes under selection in polyploids, we are in an increasingly good position to understand how polyploidy generates novelty. Here, I will review consistent effects of WGD on the biology of plants, such as an increase in cell size, increased stress tolerance and more. I will discuss how a change in something as fundamental as cell size can challenge the function of some cell types in particular. I will also discuss what we have learned about the short- to medium-term evolutionary response to WGD. It is now clear that some of this evolutionary response may 'lock in' traits that happen to be beneficial, while in other cases, it might be more of an 'emergency response' to work around physiological changes that are either deleterious, or cannot be undone in the polyploid context. Yet, other traits may return rapidly to a diploid-like state. Polyploids may, by re-jigging many inter-related processes, find a new, conditionally adaptive, normal.

## 1. Introduction

Whole-genome duplication (WGD) has wide-ranging effects on the biology of organisms, some conditionally beneficial, others deleterious [1–7]. Yet, most, if not all, plant lineages, and many animal, fungal and protist lineages have genome duplication events in their history, suggesting initial challenges can be compensated for, or that the benefits can sometimes outweigh the costs [1,4,5,8]. On shorter evolutionary timescales, WGD has been implicated in habitat adaptation and speciation. In some cases, but certainly not all, polyploids occur in expanded or distinct habitats relative to their diploid progenitors [5,9–13]. Globally, there is a trend that polyploid incidence increases with latitude [14], and polyploidy is associated with times of climatic change [6,15]. There is also abundant agricultural interest in polyploidy, as it has been linked to desirable traits such as improved stress tolerance, larger seeds or fruits or altered content of metabolites. These trends support the idea that polyploidy is commonly associated with novelty. But what are the core effects of WGD on the biology of organisms? Are there common themes or is every polyploid a unique evolutionary experiment? Do the direct effects of WGD trigger a longer-term re-tuning of the physiology of the organism? Is polyploidy something we can make use of in agriculture, and if so, what are the effects we can expect in both the short and longer term?

One of the most consistent effects of WGD is an increase in cell size [4,16,17]. When something so fundamental changes, it could impact virtually all cellular processes (e.g. [16–18]). Some effects of cell size change are probably mild, just a shudder through the system, while others are more seismic and necessitate an evolutionary response. That cell size changes associated with ploidy shifts have substantial and diverse effects on the biology of cells and organisms has been previously pointed out (e.g. [16,19]) and was recently re-addressed and synthesized

in an informative review [16]. In 1971, Bennett used the term 'nucleotype' to describe those 'conditions of the nucleus that affect the phenotype independently of the informational content of the DNA' [19, p. 296]. While the concept of cellular size and other ploidy-associated effects being independent of the information content of the DNA is very valuable, I do not adopt the term 'nucleotype' here because I do not wish to imply that the phenotypes discussed here are necessarily caused by the nucleus *per se*. That the cell size changes associated with WGD can have important evolutionary consequences has also been noted. For example, Levin [4] pointed out that the biochemical, physiological and developmental shifts associated with WGD could 'propel a population to a new adaptive sphere' [4, p. 1]. We are in an increasingly good position to understand both what this 'new normal' looks like in individual cases, and how polyploids get there, and that is the main focus of my review.

Polyploids come in two major types, auto- and allo-polyploids, which form the extremes of what is probably a continuum [3,5,20]. Autopolyploids form from within-species WGD, while allo-polyploids form from hybridization events followed by WGD, or from the fusion of unreduced gametes from genetically distinct parents [2,8]. While autopolyploids were originally thought to be rare, it is now clear that at least in plants they are quite common [3,5,21,22]. Here, I will focus exclusively on autopolyploids, what has been called 'pure polyploidy' [23], because my main interest is to discuss effects of genome duplication *per se*, and the evolutionary response to it, without the confounding effects of hybridity. There is now a rich literature available on the immediate effects of WGD and also a number of studies that compare established polyploids with their nearest diploid relatives. Both approaches have strengths and drawbacks: from studies of neopolyploids, we can discern immediate effects of WGD, but cannot know their ultimate evolutionary fate, while from studies of established tetraploids, we cannot be sure whether the differences observed arose from WGD itself, or later 'conventional' adaptive evolution (conversely, we cannot be sure that things that do *not* differ between diploids and evolved tetraploids were not at some point important WGD challenges that were rapidly contended with). As has been pointed out (e.g. [3,24]), by comparing diploids with both newly generated autotetraploids and evolved autotetraploids, we can gain a more complete picture of what the pros and cons of polyploidy are, and what cellular and developmental adjustments may arise in response. Even though I focus on plants, the broader concepts I will discuss will apply across kingdoms.

Here, following a brief overview of traits that have been repeatedly observed as responses to WGD in plants, I will discuss a small selection of traits in more detail, primarily focusing on cell size effects on stomata, pollen tubes, and the vasculature, and on cell growth and stress tolerance. In order to help us understand both the key challenges and the mechanistic basis of evolved adaptations to polyploidy, I will focus on three different autopolyploid plant systems in which adaptation to WGD has been analysed. These are: *Arabidopsis arenosa*, in which adaptation to genome duplication was studied using genome-wide outlier scans comparing natural diploids and their established autotetraploid derivatives (which have a monophyletic origin about 30 000 generations ago [25]) using several distinct metrics related to genetic diversity, differentiation and the site frequency spectrum [26–28]. The genomes of two additional species,

*Cochlearia officinalis* and *Cardamine amara*, have also been scanned for genome-wide differentiation outliers between ploidies using several differentiation statistics [29,30].

There are many important ploidy-relevant processes that I will not discuss (e.g. meiosis, chromatin remodelling, pathogen resistance, root systems, ion uptake, etc.); this is not a value judgement. The main point here is that many phenotypes are interlinked, many observed changes may be pleiotropic effects of the same underlying perturbations, and much of polyploid adaptation probably involves a global re-tuning of cellular and organismal homeostasis and cellular processes to a 'new normal'.

## 2. Common phenotypic changes associated with independent genome duplications

I have surveyed 88 studies of 67 species: 46 neopolyploids and 23 established natural autotetraploids (electronic supplementary material, table S1; summarized in table 1). From these studies, it is clear there are phenotypes common to many WGD events. An overview of the types of traits that can be affected by polyploidy is given in figure 1. One of the most frequently reported traits is an increase in cell size, particularly stomatal guard cell size (table 1; electronic supplementary material, table S1). Cell size has long been recognized to scale with DNA content and ploidy [4,17,24,31–33], so this is unsurprising. For recent discussions of hypotheses about why cell size might increase with DNA content see [18,31–34]. There is nevertheless variation in the extent to which this is true among and within species, and especially among cell types within an organism [16,34–37].

Photosynthetic rate is frequently seen to change after WGD, commonly (but not always) increasing in polyploids relative to diploids (table 1; electronic supplementary material, table S1). One mechanism that might improve photosynthetic rates is increased gas exchange through larger stomata [38–40], and some polyploids indeed have higher gas exchange rates (electronic supplementary material, table S1). Photosynthetic rate per leaf area shows a variable response to WGD, probably owing to variation in leaf morphology, but chloroplast numbers per cell (and photosynthetic rate per cell) reliably increase with cell size in polyploids, while photosynthetic rate per chloroplast remains unchanged [41–43].

Increased cell size may have additional context-specific consequences such as the relationship of legumes with their nitrogen-fixing symbiotic partners. For example, polyploids gain more vigour from mutualism with a wide range of *Sinorhizobium* strains than diploids [44–46]. For autotetraploid *Medicago sativa*, this is mediated by larger nodules with larger nitrogen (N)-fixation zones, which leads to greater N-fixation. The polyploid advantage in N-fixation could stem from multiple effects including larger cell size, which correlates with reduced $O_2$ diffusion rates that would provide a more efficient environment for N-fixation (an anaerobic process) [47]. For more on polyploidy and its effects on biotic interactions see [47–50].

A commonly reported feature of polyploids is their stress tolerance [4,51–53]. One of the most repeatable tolerances for both neo- and established polyploids is higher drought tolerance. Improved salt tolerance has also been reported for many neopolyploids (table 1; electronic supplementary material, table S1). Other stress resistances are more variable,

**Table 1.** Major traits repeatedly reported in studies comparing autopolyploids to their diploid progenitors or closest relatives (summary version of the electronic supplementary material, table S1). (A more detailed version of this table, including additional traits, species names and descriptions of the type of each polyploid, are given in the electronic supplementary material, table S1. 'Rel to 2X' indicates the type of difference relative to the diploid comparand, where 'no diff' means no significant difference. 'Reports' refers to the number of studies within the 88 included that report these differences for each type of polyploid (note that this not an exhaustive list). 'Type' refers to the type of polyploid, where neo means newly generated either in the laboratory by colchicine or oryzalin, or naturally (spontaneous), while established (evolved) refers to natural polyploid taxa found in nature of usually uncertain age, but clearly established as a distinct entity in native habitats.)

| trait class | trait | rel to 2X | reports | type |
|---|---|---|---|---|
| cell size | leaf cell size | larger | 3 | neo (colchicine) |
| | | | 5 | neo (natural, cultivar) |
| | | | 4 | established (evolved) |
| | guard cell size | larger | 14 | neo (colchicine, etc.) |
| | | | 4 | neo (natural, cultivar) |
| | | | 6 | established (evolved) |
| | | no diff | 2 | neo (oryzalin) |
| gas exchange | stomatal density | lower | 9 | neo (colchicine, etc.) |
| | | | 4 | neo (natural, cultivar) |
| | | | 6 | established (evolved) |
| | | no diff | 2 | neo (oryzalin) |
| | stomatal conductance | higher | 6 | neo (colchicine) |
| | | | 2 | neo (natural, cultivar) |
| | | | 6 | established (evolved) |
| | | lower | 3 | neo (natural, cultivar) |
| | | no diff | 2 | neo (colchicine, oryzalin) |
| photosynth. | photosynthetic rate | no diff | 2 | neo (colchicine) |
| | | | 1 | neo (natural) |
| | | | 1 | established (evolved) |
| | | higher | 10 | neo (colchicine) |
| | | | 3 | established (evolved) |
| | | lower | 2 | neo (colchicine) |
| | chlorophyll content | higher | 10 | neo (colchicine) |
| | | | 1 | established (evolved) |
| | | no diff | 4 | neo (colchicine, oryzalin) |
| stress | drought tolerance | higher | 10 | neo (colchicine) |
| | | | 2 | neo (natural, cultivar) |
| | | | 5 | established (evolved) |
| | salt tolerance | higher | 8 | neo (colchicine) |
| | | | 2 | neo (natural, cultivar) |
| | | lower | 1 | neo (natural, cultivar) |
| | | | 1 | established (natural) |
| | | variable | 2 | neo (natural) |
| | reactive oxygen species | higher | 2 | neo (colchicine, oryzalin) |
| | scavenging | | 2 | established (evolved) |
| | anti-ox activity | higher | 5 | neo (colchicine, oryzalin) |
| hydraulics | xylem diameter | no diff | 1 | neo (oryzalin) |
| | | higher | 2 | neo (colchicine) |
| | | | 3 | established (evolved) |
| | hydraulic conductivity | lower | 1 | established (evolved) |
| | | higher | 2 | neo (colchicine) |
| | | | 1 | established (evolved) |
| | cavitation | no diff | 1 | neo (colchicine) |
| | resistance | | 2 | established (evolved) |

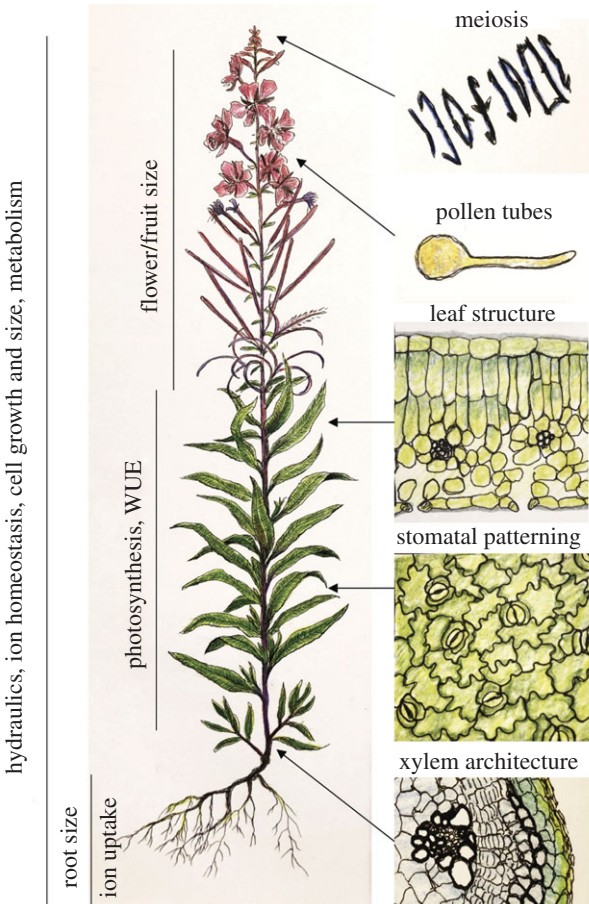

**Figure 1.** Summary of some of the major changes associated with WGD in plants. Some traits like ion homeostasis, cell growth and size, and metabolism are whole-plant phenotyes in that any or all cells could be affected. Others are more specific to certain tissues. The traits on the right show those that are more cell or tissue-specific, but often have equally global effects for the plant (illustrations, K.B.). WUE, water use efficiency. (Online version in colour.)

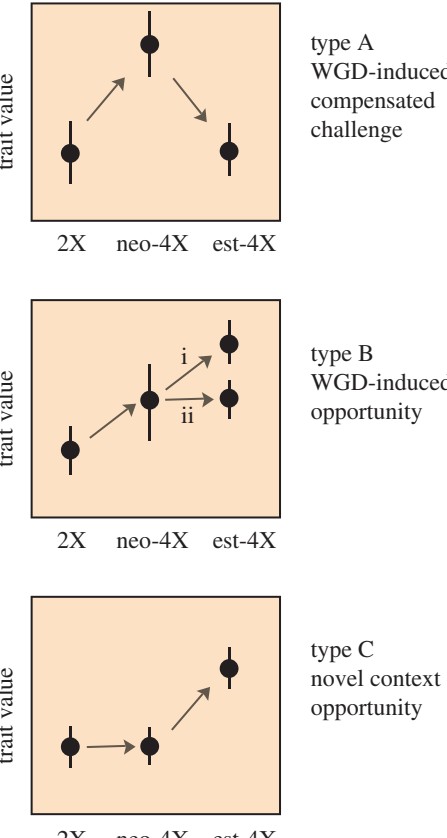

**Figure 2.** Different outcomes for traits after WGD. 'Type A' traits are those which change upon WGD, but then evolve back towards the diploid state in evolved tetraploids. These would be traits that would represent challenges that are faced after WGD but that would not be identified as challenges if only neotetraploids are studied, and not understood to have been challenges that necessitated evolutionary responses if only evolved tetraploids are compared to diploid progenitors. 'Type B' traits also change upon WGD, but then are either further modified (i) or maintained (ii) in the evolved tetraploids. These can be thought of as 'opportunity' traits. 'Type C' represents traits that do not change upon WGD, but where polyploidy provides a context in which novelty in this trait is advantageous or facilitated in the mid to long term. (Online version in colour.)

for example, heat and cold both have examples of higher and lower resistance among polyploids, and there are examples of elevated UV tolerance, low hydric tolerance and low freezing tolerance (electronic supplementary material, table S1). Other traits are also listed in the electronic supplementary material, table S1.

Interestingly, while both new and established tetraploids commonly have increased cell size, there is also evidence that over evolutionary time, cell size tends to evolve back downwards [41,54]. Whether this is more attributable to a decrease in DNA content over time [5,55], or other factors, is not yet clear. My suspicion is that while genome downsizing (which could be adaptive or selectively neutral) probably plays a role, increased cell size may also present cell-type-specific challenges for polyploids (e.g. to do with stomatal opening, pollen tube growth and vasculature), and these could drive selection for genetic changes that again reduce cell size, or compensate for the increase, as discussed below.

## (a) Types of responses to whole-genome duplication fall into several classes

It is becoming evident that some traits which change in response to WGD in many independent cases are maintained in established polyploid lineages, while others are apparently not. These trends become evident when we compare diploids, neotetraploids and derived tetraploids (figure 2). I grouped what I see as the three major trends into types. 'Type A' traits are those where a WGD-induced change is followed by an evolutionary response that returns the trait partly or wholly to a diploid-like value. In such cases, immediate effects of polyploidy probably have costs, such that selection will favour new variants that undo or work around them (figure 2). These traits could be subtle or invisible in studies that compare only diploids and evolved tetraploids, while in studies of only neotetraploids, we would not necessarily know that they represent challenges that will quickly disappear. Some possible examples I will discuss below include leaf cell size, pollen tube growth rate, cell growth rate and perhaps salt tolerance. 'Type B' traits are those where WGD induces a change that is either maintained or further elaborated by evolution. These traits could be discovered in both types of comparison noted above. Some examples discussed below might include pollen tube width, stomatal density and drought tolerance. Both A and B type changes may facilitate, or even necessitate, evolutionary responses in other traits or processes. 'Type C' traits would appear not as immediate consequences, but sometime

after WGD. Where type C traits appear repeatedly in independent polyploids, we could hypothesize they might arise as parallel evolutionary responses to consistent immediate or longer-term effects of WGD.

## 3. Cell size

While the cell size changes that occur in polyploids probably have global effects on the organism [16,18,33], I will focus here on three cell types in plants where cell size changes may induce significant physiological shifts: guard cells, pollen tubes and xylem.

### (a) Guard cell dynamics and development

An increase in guard cell size may have tremendous impacts on the biology of a plant. Guard cells are epidermal cells that flank stomata, the pores in the leaf surface that allow gas exchange and are thus essential for energy production via photosynthesis [56,57]. Guard cells are very commonly larger in both new and evolved polyploids (table 1; electronic supplementary material, table S1), though there is evidence from comparing neo- and established tetraploids of several species that while there is a large shift in guard cell size triggered by WGD, established polyploids exhibit guard cells of intermediate size, indicating a reduction over time [41,54], suggesting this may in part be a 'type A' trait (figure 2). Stomatal density consistently decreases in polyploids (electronic supplementary material, table S1). As there is a strong developmentally regulated correlation between larger stomatal size and lower density [39], this trend is expected and suggests this regulatory relationship remains unperturbed after WGD (table 1; electronic supplementary material, table S1).

Stomatal size and density affect both the rate of $CO_2$ uptake into the plant (and thus the photosynthetic rate) as well as the transpiration rate of water [56]. This is highlighted, for example, in *Arabidopsis thaliana*, where mutants with higher stomatal density have higher $CO_2$ assimilation under high light [58], and plants with reduced stomatal density have reduced gas exchange and photosynthetic rates [59]. Because polyploids have both larger stomata and a lower density, it is not immediately obvious what to expect. Depending on the rate and extent (aperture) of responsiveness of stomata to environmental or hormonal cues, polyploids could have either higher or lower gas exchange rates. Indeed, both lower and higher gas exchange, $CO_2$ assimilation and photosynthetic rates have been reported in polyploids (electronic supplementary material, table S1).

A critical feature of guard cells is the rate at which they can respond to both external or internal cues, and cell size may have a direct physical effect on this [18,40,60,61]. The opening and closing of stomata occur via turgor changes that are achieved by shunting ions through plasma membrane channels between guard cells and adjacent subsidiary cells, such that water follows by osmosis and adjusts the turgor of the cells [62]. Smaller guard cells may be able to open and close stomata more rapidly owing to the lower ion flux needed to generate turgor changes [18,39,60,63]. However, inefficiencies associated with guard cell geometry or density can also be accommodated by adjusting the speed of opening or closing [38,40] or the stomatal aperture [63]. A rapid response to environmental conditions maximizes the extent to which a plant can take advantage of $CO_2$ and sunlight to make

sugars by photosynthesis, while allowing it to respond efficiently to changes in transpiration rate [38,60]. This is clearly important in nature. For example, a survey of *Banksia* species showed that leaves adapted for higher gas exchange have smaller stomata with faster responses, which offset dehydration risks [38]. Larger and slower stomata are associated with species growing in wetter, shadier conditions [64]. Thus, we might expect that neopolyploids, with their larger guard cells, have more sluggish responses owing to the challenges of achieving the same osmotic potential in larger cells. In many habitats, this slowed response rate could be maladaptive, necessitating, for example, the evolution of more efficient ion pumps to adjust opening/closing rates. For this reason, stomatal opening/closing rates can be considered a 'type A' trait (figure 2) in which a challenge that arises as a direct consequence of the cell size changes associated with WGD is compensated for by adaptive evolution.

Interestingly, in the three genome analyses of adaptation to polyploidy available to date in plants, in *Ar. arenosa* [26,28], *Ca. amara* [30] and *Co. officinalis* [29], there is evidence that guard cell responsiveness and patterning may be under selection. Ion fluxes between guard cells and adjacent subsidiary cells, mediated by plasma membrane ion transport, are critical for guard cell turgor changes (e.g. [62,65]). Key components for generating ion fluxes throughout the plant, including guard cells, are plasma membrane localized $H^+$-ATPases, which power the flux of $K^+$ and other ions by generating proton gradients using ATP [66]. Increasing the efficiency of pumps has been proposed as a way to get faster stomatal responses [60]. In this light, it is interesting that the main guard cell $H^+$-ATPase, AHA1, shows evidence of having been modified by selection in established tetraploids of *Ar. arenosa* and also shows evidence of ploidy-associated differentiation in *Co. officinalis* [26,28,29]. Although AHA1 has diverse functions throughout the plant (e.g. [67–71]), it is very important in guard cell opening (and closing) [65,69,72–74]. While AHA1 does not seem to be under selection in *Ca. amara*, ARR1, which regulates AHA1 and its close homologue AHA2, is ploidy-differentiated [30,68]. In addition, all three species show evidence of selection having targeted regulators of AHA1: in *Ar. arenosa*, PPI2 and in *Co. officinalis*, PPI1 are more differentiated between diploids and tetraploids than the genome average [29,30]. These proteins bind to and stimulate AHA1 [75,76], though their *in planta* functions are still largely mysterious. Another very interesting differentiation outlier in *Ar. arenosa* is the AHA1-regulator PATROL1 [30]. In *Ar. thaliana*, PATROL1 controls dynamic localization of AHA1 to the plasma membrane in guard and subsidiary cells, and stomatal opening requires the interaction of PATROL1 and AHA1 [77,78]. Overexpression of PATROL1 in *Ar. thaliana* causes plants to have a faster stomatal response rate to light and other cues, and a larger aperture, coupled with higher photosynthetic rate and plant growth [78,79]. Mutants have reduced guard cell responsiveness to environmental cues [78]. Another intriguing protein related to guard cell opening rates that also shows good evidence of selection in *Ar. arenosa* and differentiated by ploidy in *Co. officinalis*, is α-amylase-like 3 (AMY3) [26,29]. This enzyme is chloroplast-localized, and aids in the rapid turnover of starch in the morning to facilitate rapid stomatal opening downstream of AHA1 [80,81]. Thus, regulation of AHA1 and stomatal dynamics are tightly tied with environmental cues, and the hints that at least twice independently, selection may have targeted the gene that

encodes AHA1, might suggest mechanisms by which polyploids can adapt their stomatal opening and closing rates to cell size changes.

The hormone abscisic acid (ABA), which is produced primarily in roots and transported to shoots, triggers stomatal closure [62]. The ABA-responsiveness of particular genotypes can affect the responsiveness of stomata in polyploids. For example, polyploid *Ar. thaliana* (Col-0 strain) has enhanced closure of stomata relative to diploids, which correlates with reduced transpiration rates and increased drought and salt tolerance [37]. However, another tetraploid strain, Me-0, does not show this effect. Me-0 has lower ABA-responsiveness than tetraploid Col-0, leading to slower stomatal closure and elevated gas exchange [82], suggesting ABA responses play a role in ploidy-induced changes to guard cell function. A possible mechanistic link to ABA comes from the $Ca^{2+}$-ATPase ACA8 [30], which is a plasma membrane localized $Ca^{2+}$ efflux channel [83,84] and shows evidence of selection in *Ar. arenosa*. While in *Ar. thaliana*, ACA8 has effects on many traits throughout the plant [85,86], it also plays a role in the closure of stomata, specifically the stomatal responsiveness to both $Ca^{2+}$ and ABA [86]. The calmodulin-binding protein IQM1 is another interesting genome-wide outlier for differentiation in *Ar. arenosa* [30]. IQM1 is important for the responsiveness of stomata to ABA and other cues, and affects stomatal aperture [87].

Another way in which gas exchange can be adjusted is by altering the patterning of stomata on leaf surfaces via changes in a number of different genes [57]. Over evolutionary time, while cell size correlates well with genome content, stomatal density is less strongly correlated, suggesting the size/density ratio of stomata can be adjusted by a locally adaptive evolutionary response [88]. In the context of polyploid adaptation, it is interesting that several genes implicated in stomatal precursor cell divisions, and ultimate stomatal patterning, show evidence of being genome-wide outliers for metrics indicative of selection and differentiation in *Ar. arenosa*, and *Co. officinalis*, [29]. The protraction of the divergence of the size/density ratio over evolutionary time suggests this might be a 'type C' trait (figure 2), to adjust gas exchange to suit local habitats in a polyploid context.

## (b) Ion regulation and cell (and plant) growth rate

Reduced growth rates in polyploids have been reported numerous times, but in almost all cases, these reports come from neopolyploids (electronic supplementary material, table S1). Interestingly, in one case in which both a neo- and established polyploid were compared to a diploid (*Co. officinalis*), both had reduced growth rates relative to the diploid, but the difference was greater for the neotetraploid, suggesting the growth rate reduction associated with WGD was compensated at least to some extent over time [55], suggesting growth rate is also a 'type A' trait (figure 2). Why growth rates are slower in polyploids is not entirely clear. While DNA content in diploids often correlates with cell division rates (e.g. [89]), it does not seem that WGD substantially affects the duration of the mitotic cell cycle [90,91]. Reduced growth may instead be related to changes in cellular metabolism, or hormone levels, both of which are known to change after WGD (electronic supplementary material, table S1). For example, in autotetraploid *Brassica rapa* (pak choi type) and apple, lower growth rates correlate with reduced

levels of the hormone auxin (and in the case of apple, also Brassinolide) [92,93]. However, in another study, in *Ar. thaliana*, cellular growth rate was not affected by ploidy, but polyploids had increased expression relative to DNA content of genes related to cell wall production [34], an observation which is frequently associated with accelerated growth.

A major way in which plant cells grow is by auxin-triggered acidification of the apoplast and cell walls by plasma membrane localized $H^+$-ATPases. The auxin-triggered proton efflux in turn activates pH-sensitive expansins that loosen cell walls to allow turgor-driven growth [68,70,94,95]. Might it be that larger cells, with their greater ratio of surface area to volume, sometimes face challenges in cell wall acidification and thus experience slowed growth? Alternatively, could improving acidification of cell walls be a way that polyploids can, over evolutionary time, overcome growth hindrances they experience early on for other reasons? There is clear evidence that proton pumps play a role in cell growth: in *Ar. thaliana*, a double heterozygote of AHA1 and its closest paralogue, AHA2 (double homozygotes are lethal), shows a reduced ability to acidify the apoplast [68]. However, AHA1 is one of the most widely expressed and multifunctional of the plasma membrane $H^+$-ATPases in *Ar. thaliana* [67], so, as for many of the genes that may have been under selection after WGD, this raises the question what AHA1 might actually be under selection for. Are AHA1 and its regulators targeted because of their effects on guard cell responsiveness, with pleiotropic effects on cell growth and other processes, or *vice versa*, or something else entirely?

In the context of cell growth, another intriguing set of genes that show evidence of differentiation and/or selection in the tetraploids are several channels involved in pH maintenance of the trans-Golgi network and endoplasmic reticulum. This pH maintenance results from collaboration of $H^+$ pumps and $K^+$ channels, and is important for cell growth and cell wall deposition, among many other things [96]. Among the genes probably under selection in *Ar. arenosa* and showing ploidy differentiation also in *Co. officinalis* are several $Na^+/H^+$ antiporters (NHX) channels, which are important for mediating pH, cell turgor, stomatal opening and cell growth [97]. NHX5, for example, which shows ploidy differentiation in *Ar. arenosa* [30], is involved, together with NHX6, in regulating endomembrane system pH, particularly of the trans-Golgi network; mutants have reduced cell wall deposition, slower tip growth and are sensitive to salt [98,99]. Many $K^+$ channels are located in the trans-Golgi network (e.g. KCO6, KEA5 and KEA6) or endoplasmic reticulum (e.g. KUP9) [100–102]. KUP9 shows ploidy differentiation in both *Ar. arenosa* and *Co. officinalis* [29,30], and KCO6, KEA5 and KEA6 in *Ar. arenosa* [26,30]. In *Ca. amara*, there is evidence of ploidy differentiation in a proton pump, VHA-a1 [30], which is also targeted to the trans-Golgi network, and is important for secretory and endosome trafficking, as well as cell wall synthesis [96,103,104]. All of these proteins are involved in general ion homeostasis in the cell, and thus will have effects throughout the plant, but the possible role of modifications to these systems in altering dynamics of cell growth rate in polyploids over evolutionary time merits exploration.

## (c) Pollen tube tip growth

Another major cell-size-related challenge to polyploids may be tip growth in pollen tubes, upon which the fertility of

sexually reproducing plants depends. How does a ploidy-associated increase in cell size affect pollen tubes? Pollen tubes in neopolyploids, where studied, are wider and grow slower, probably because wider tubes require more cell wall material to be deposited [105]. However, pollen tubes in *established* autopolyploids grow as fast as, or in many cases faster than, those in diploids. In a large survey of 451 species, very young (intraspecific) polyploids had large and slow pollen tubes, while more established polyploids had fast pollen tube growth rates, equalling or exceeding their diploid relatives [106]. These observations suggest that over time, genetic effects can overcome the immediate slowing effects arising from WGD [105,106]. It seems that pollen tube growth rate is a 'type A' trait (figure 1) where the slow-down that arises from genome duplication is reverted over evolutionary time; but this challenge is solved not by undoing the initial change (pollen tube width), but via work-arounds, suggesting that pollen tube width is a 'type B' trait. For example, in *Betula* species, polyploids have a similar pollen tube growth rate as diploids, but this involves not a reduction in pollen tube size, but rather faster wall production rate and volumetric growth [105]. Similarly, in *Handroanthus*, a sexual hexaploid has a similar pollen tube growth rate to a diploid, also owing to faster wall production and volumetric growth. By contrast, an apomictic tetraploid *Handroanthus* does not show a similar pollen tube growth rate to the diploid, highlighting that it is probably selection for fertilization success that is driving the acceleration of polyploid pollen tube growth rates [105].

How might polyploids accelerate the growth of large pollen tubes? Interestingly, the three available genome scans for ploidy-associated selection and differentiation hint that the mechanisms of tip growth may be modified during the evolution of polyploid lineages. Pollen tubes grow by the transport of vesicles with cell wall and plasma membrane materials exclusively to the tip [107]. Polarized gradients of $H^+$, $Ca^{2+}$ and $Cl^-$ ions are important features of tip growth, and the pollen tube shank generates proton effluxes for which the tip acts as a sink [107–109]. These fluxes depend on plasma membrane $H^+$-ATPases (AHA6, AHA8, AHA9) [109,110]. In *Ar. arenosa*, for which we know that pollen tube growth in the established autotetraploid is as fast as, or faster than, that seen in the nearest diploid relatives (J. Westermann and K. Bomblies 2020, unpublished data), pollen tube expressed $H^+$-ATPase AHA8 has strong evidence of having been under selection in the tetraploid [26,28,30]. In *Co. officinalis*, AHA9 is differentiated between diploids and tetraploids [29]. In addition to the pollen-expressed $H^+$-ATPases, $Ca^{2+}$ signalling and the GTPase Rho family of plants (ROPs) and their targets are also critical for maintaining polarity of pollen tube growth [111]. ROPs are toggled 'on' and 'off' by activator RopGEF proteins and suppressor Rho-GTPase activating proteins (Rho-GAPs) (e.g. [112–114]). Both activities are essential for the regulation of tip growth [107,108,111]. In *Ar. arenosa*, there is evidence of ploidy differentiation in REN1 [30], which encodes a RhoGAP essential for pollen tube tip growth that functions by constraining ROP1 GTPase activity to the growing tip of the pollen tube [112]. In addition, there is strong evidence for selection having acted on a pollen tube localized kinase, AGC1.5 in tetraploid *Ar. arenosa* [26,28,30]. AGC1.5 affects pollen tube growth by controlling ROP targeting by phosphorylating RopGEF proteins [115]. In *Co. officinalis*, a RopGEF protein is differentiated between ploidies [29].

In *Ca. amara*, there is less evidence of selection on pollen tube growth [30], but some *Ca. amara* are apomictic and thus selection for pollen performance may not be as strong in this species [116]. Nevertheless, one protein differentiated between diploids and tetraploids in *Ca. amara* is VHA-a1 [30], a subunit of a Golgi-localized $H^+$-ATPase (V-ATPase) that also plays a role in pollen development [117]. Another protein differentiated between diploid and tetraploid *Ca. amara* is ACA9, a plasma membrane expressed $Ca^{2+}$-ATPase [118]. While $Ca^{2+}$ fluxes are important for almost all signalling, growth and response processes throughout the plant, unlike the other ACAs, ACA9 is pollen-tube-specific, and *aca9* mutants in *Ar. thaliana* are defective in pollen germination and show slow pollen tube growth [118].

Whether any, some or all of the proteins mentioned above are involved in increasing pollen tube growth rate in these species remains to be tested, but together with the phenotypic data from other polyploids, we now have a list of candidate proteins and processes that can be followed up.

## (d) Xylem architecture

Cell size also has interesting implications for the functionality of xylem, the water-transporting tissue in plants. Though xylem structure in polyploids has received less attention than it probably deserves, there are some studies from which we can already gain interesting insights. Several studies noted that xylem conduits are wider in polyploids, presumably owing to the overall trend towards larger cells [119–122]. What consequences might this have for the water relations of the plant? On the one hand, wider xylem elements can carry more water, meaning stomata will not need to close as soon, which could be predicted to make plants more drought-tolerant. However, wider elements, when all else is equal, are also more prone to embolism and/or implosion, which might make the plant more drought-sensitive [123–127]. On the other hand, structural changes to other features of the xylem can compensate for increased cavitation risk [128,129]. These observations would include water relations among 'type B' traits (figure 2), in that wider vessels are maintained, though their safety features are modified.

To clarify how xylem might be affected by cell size, I briefly touch on some of its basic features. Xylem cells are dead at maturity, but develop from living cells. Xylem cells (tracheids) and muticellular tubes (vessel elements) are single water-conducting units connected via small pores with membranes called pits [126,127,129]. Transpiration from leaves, through open stomata, generates negative pressure in leaves that pulls water up through the xylem. As the pull from leaves increases (e.g. as transpiration increases in hot or dry weather), or water availability from the soil declines, cavitation (formation of air pockets owing to negative pressure) can occur in the xylem cells, obstructing water flow [123,124,126,127]. Cavitation resistance is an important feature of drought-adapted plants, and there is a general (albeit weak) association across angiosperm diversity between cavitation resistance and arid habitats [130]. An important feature that promotes cavitation resistance is the structure of the pits connecting xylem cells, which are critical for containing embolism, as they prevent air pockets from expanding from cell to cell [126,128]. However, there is a trade-off; morphological features of pits that improve safety, also reduce water conductivity [126,129,131]. As noted above, wider elements, as polyploids

seem to generally have, should increase water conductivity, allowing stomata to stay open longer if water is in sufficient supply. However, plants with wider xylem elements may also pay for that increased conductivity with elevated cavitation risk. Yet, over evolutionary time, xylem element width is not always associated with increased cavitation risk, and there is only a weak correlation between conduit diameter and drought-induced cavitation risk. Interestingly, there is a much stronger negative correlation between the width of conduits and freezing tolerance, which suggests freezing may be a more important driver of selection on conduit diameter than drought [126]. This is interesting in the light of the observation that freezing tolerance in two cases where it was studied in polyploids is indeed lower than in related diploids [132,133].

In *Chamerion angustifolium*, vessel elements of both neo- and evolved tetraploids are wider than in diploids, and while tetraploids have higher hydraulic conductivity than diploids, they show no difference in vulnerability to cavitation [119]. Interestingly, conductivity is even higher in evolved tetraploids than neotetraploids (thus, a 'B-i type' trait; figure 2), though xylem element diameters and cavitation risk does not increase further. This observation suggests that conductivity can be further enhanced through other means, perhaps as an evolved adaptive response to water relations changes caused by WGD [119]. The higher conductance of the established polyploids correlated with greater drought resistance [119]. *Atriplex canescens* established autotetraploids also have wider xylem elements and are also not more cavitation-prone than diploids [121]. However, unlike *Ch. angustifolium*, *At. canescens* tetraploids have lower hydraulic conductivity than diploids in unstressed conditions, which correlates with leaf anatomical features that reduce water loss. They are, however, more resistant than diploids to drought-induced loss of conductivity, and thus can maintain higher conductance, $CO_2$ assimilation and photosynthesis under heat and water-limited conditions than diploids [121,134]. Established autotetraploid birch is similarly better able than the diploid to maintain water pressure in drought conditions, and this is again correlated with anatomical features in leaves that reduce water loss [135]. A more recent study on birch showed that polyploids, which are found in more stressful environments than diploids, have both higher vessel diameters and higher resistance to drought-induced embolism [122]. The authors go on to demonstrate that this higher resistance comes from differences in pit architecture that confer greater hydraulic safety [122].

In polyploids, if xylem 'safety features' can be adjusted to offset cavitation risk from wider cells, they could potentially maintain higher water conductivity, allowing them to transpire at the same rate as diploids at less negative water potentials, and thus lower risk of cavitation [119,127,136]. Indeed, cavitation risk, where it has been measured, does not seem to increase with WGD [119–121], suggesting it may be a plastic feature sensitive to changes in water relations, rather than a product of longer-term adaptive evolution. Can xylem have such plastic adaptive responses? Interestingly, there is evidence that there can indeed be plastic developmental responses in xylem that can mitigate challenges. In sequoia trees, for example, with increasing height on single trees, there is an increase in cavitation resistance correlated with decreased pit apertures [131], showing that at least some safety features can arise as plastic adjustments within a single plant. In *Ilex aquifolium*, single trees can modify growth over time with changes in climate, adjusting xylem element number and clustering, features which correlate positively with conductivity and negatively with embolism risk [137]. Interestingly, polyploid *Capsicum annuum*, which has wider xylem elements, also had more xylem elements in the roots, suggesting this might also be a plastic trait that offsets increased cavitation risk by increasing redundancy [120]. These findings suggest that plastic responses could, to some extent, offset other changes that occur in response to WGD, while other features (like the increased conductance seen in *Ch. angustifolium* established tetraploids [119]), may arise later, suggesting these are 'type C' traits (figure 2).

## 4. Stress resilience

The high drought and salinity tolerance of virtually all tested neopolyploids (electronic supplementary material, table S1) is intriguing as it suggests this arises from features that most neopolyploids share. At least some of these traits seem to persist in polyploids and can affect habitat specialization in some cases. Some polyploids are found in drier, more ruderal or more extreme habitats than diploids (e.g. [138–143]). This may have important implications also in the context of climate change. For example, testing perennial ryegrass diploids and tetraploids against both current and future climate scenarios predicted for England in 2080 (with more drought), polyploids fared better in the drier future conditions, while diploids outcompeted polyploids in wetter present regimes [144]. On the other hand, habitat differentiation is negligible or absent in other examples [5,9,145].

From the literature currently available, it seems that drought tolerance is commonly reported for both neo- and established polyploids, suggesting it is maintained for extended time after WGD and thus a 'type B' trait (figure 2). For reports of salinity tolerance, on the other hand, there is a strong bias towards neopolyploids (table 1; electronic supplementary material, table S1). The sparsity of salinity tolerance examples among studies of established polyploids may be a sampling artefact, or it could be that in fact the salinity tolerance that arises from WGD is not maintained as often over evolutionary time (that is, it is a compensated 'type A' trait; figure 2). But why? What might be distinct about salinity and drought tolerance of polyploids?

Stress-tolerant polyploids commonly show a qualitatively similar, but considerably milder, stress response than diploids. Changes in, for example, gas exchange, stomatal opening, chlorophyll content, photosynthesis, water content, etc., generally occur in both diploids and polyploids in response to drought or salinity, but are less severe in polyploids (e.g. [134,135,146–149]). Why? In some studies, the differences observed can be attributed to differences in leaf anatomy and/or transpiration rate between the diploid and tetraploid (e.g. [37,147]). For example, mutant and overexpression studies in *Arabidopsis*, rice, maize and barley, all support the notion that stomatal size and density alone can account for differences in transpiration rate, drought tolerance and water use efficiency [150–154]. What is perhaps a bit less intuitive, is that stomatal size and density may also directly impact salinity tolerance, probably by altering water transport in the plant (and thus $Na^+$ ion uptake). For example, in strawberry

and quinoa, salt tolerance correlates with low stomatal density; thus, salt tolerance in these species is thought to be owing to the low transpirational flux, and thus slowed uptake of toxic ions [155,156]. Salt tolerance can also be related to stomatal closing, based on the observation that a diploid mutant in rice that has increased stomatal closure and reduced stomatal density (like most polyploids), is also highly drought and salt tolerant [157].

In other studies, there is evidence that polyploids suffer less cellular stress and have higher levels of reactive oxygen species-scavenging and anti-oxidant capacities that could lead to generalized stress resilience [158–161]. Lower resource availability under stress conditions may also in some cases favour polyploids because they are commonly slower growing, causing them to suffer less cell damage [160]. There are also hints that the 'stress hormone' ABA plays a role in at least some cases: tetraploid pak choi, for example, has higher ABA levels [92], while in Rangpur lime, tetraploid rootstocks had higher ABA levels and higher constitutive expression of drought response genes than diploid ones, and tetraploid roots confer greater drought tolerance to the shoot even when grafted with diploid scions [162].

There is substantial overlap between salinity tolerance, and ion homeostasis and uptake. For example, salt tolerance in tetraploid B. rapa and Ar. thaliana correlates with constitutively higher $K^+$ levels in the shoot, and maintenance of a higher $K^+/Na^+$ ratio in saline conditions [161,163]. In the natural polyploid, Robinia pseudoacacia, salinity treatment caused $K^+/Na^+$ ratios to decrease in the diploid, but not the tetraploid [164]. Maintaining high $K^+/Na^+$ ratios is critical for plant survival as $K^+$ is essential for plant life, while $Na^+$, being chemically similar, cannot be as effectively excluded when $K^+$ becomes limiting or $Na^+$ is present in excess [165,166]. Interestingly, several genes involved in maintaining $K^+/Na^+$ ratios (SOS1/NHX7, HKT1) [97,167–169], and other ion homeostasis regulatory channels implicated in salt and other stress responses (CCC1, KEA2, KEA3) [170,171], as well as a number of other genes that have been associated with salt tolerance, are ploidy-differentiated (and in Ar. arenosa, show additional signatures suggestive of selection) in the genome scans for adaptation to autopolyploidy published to date [26,29,30]. Links to global ion homeostasis are evident, for example, from the observation that a mutant for the trans-Golgi-network-localized NHX5 channel (ploidy-differentiated in Ar. arenosa [30]), affects cell growth and is salt sensitive [99]. As noted above, several additional trans-Golgi-network-localized $K^+$ channels also show evidence of selection and/or differentiation in tetraploids (e.g. KEA5, KEA6, KUP7) [26,29,30]. KEA5 and KEA6 work together with NHX5 in endosomal pH control [102]. An increased ability to maintain a high $K^+/Na^+$ ratio is probably key in making polyploids more salt tolerant. What is currently not clear, however, is what it is about WGD that allows them to do this, and why associated genes might be under selection in tetraploids after WGD, including in the two species that are not found in saline habitats (Ar. arenosa and Ca. amara). Probably , there is a fundamental feature of $K^+$ homeostasis in polyploids that necessitates this evolutionary response, where the initial WGD-associated perturbation has spillover effects on salt tolerance. Then the question arises, if the evolutionary response to this challenge (if it is one) 'locks in' or suppresses the salinity tolerance that these tetraploids presumably acquired as a consequence of WGD. In one case, Co. officinalis, salinity tolerance in the evolved tetraploid seems to be lower than the diploid, though the tetraploid is found in more saline habitats [29]. Together, these observations hint at a link between salinity tolerance and an overall perturbation of $K^+$ homeostasis in neotetraploids that necessitates adaptive evolution, but what the fate of the (probably incidental) salt tolerance is in this adaptive process is not yet clear.

## 5. Conclusion

Effects of WGD on cell biology, and the evolutionary response to it, are complex, but provide exciting new questions that have the potential to provide novel insights into fundamental aspects of biology. While each polyploid will, to some extent, be its own unique evolutionary experiment, it is also becoming clear that there are both phenotypic and genetic commonalities across different examples that probably point back to the influence of WGD on core functionalities like ion homeostasis, cell size and cell growth regulation, as well as their interconnected effects. It appears from the survey presented here, that core physiological traits central to cellular biology are commonly 'type A' traits, in which WGD causes an initial challenge, which can either be undone, or to which solutions or workarounds subsequently evolve. Now that we also know something about not only the phenotypes associated with both new and established polyploids, but also the genes that might be under selection to get them there, we are in a position to generate testable hypotheses. There is also ample scope for more explicit comparative studies, which will gain power from being coupled with physiological understanding of the underlying traits. For example, it seems from this survey that drought tolerance is associated with both new and established tetraploids (and thus may be a retained 'type B' trait), while salt tolerance is almost exclusively reported in neotetraploids (suggesting it may be a side effect of a WGD-associated challenge that is compensated for, and thus an 'A-type' trait). Is it just that salt tolerance has not been sufficiently tested in established tetraploids, or are there associated costs that make it more difficult to maintain than drought tolerance? From the small number of genome scans for adaptation to autoploidy published to date, we cannot yet draw sweeping generalities, but one thing that is striking already is how many proteins with 'plant-wide' core cellular functions show evidence of selection (e.g. AHA1, ACA8, NHX5 and many others, like condensins, mediator and subunits of PolII that I did not discuss). Are such genes under selection to correct cellular challenges that affect the entire plant, or to compensate very specific traits (like stomatal closure rates)? Does modification of the encoded proteins have pleiotropic effects throughout the plant? It may be that over longer evolutionary timespans, spillover effects would be locked in if they happen to be beneficial (type B), or be compensated for if they are not (type A). In the shorter term, emergency responses probably take precedence, even if they inadvertently impact other processes in the plant, and thus lead to an organismal phenotypic shift that is only in part directly adaptive. Thus tetraploids, in which many aspects of their fundamental cell biology shift all at once, may need to go through a process of evolutionary 're-jigging' that may create something truly new as they find their new normal.

Data accessibility. This article has no additional data.

**Competing interests.** I declare I have no competing interests

**Funding.** My work on polyploidy is supported by an ERC consolidator grant (no. CoG EVO-MEIO 681946).

**Acknowledgements.** I would like to thank Christine Faulkner (JIC, Norwich, UK) for very helpful discussion and comments on the manuscript, and members of my laboratory for helpful discussions and enthusiasm.

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
