## [Reviewer comments · Proceedings of the Royal Society B: Biological Sciences]

Review History

RSPB-2020-2154.R0 (Original submission)

Review form: Reviewer 1

Recommendation

Accept with minor revision (please list in comments)

Scientific importance: Is the manuscript an original and important contribution to its field?

Excellent

General interest: Is the paper of sufficient general interest?

Excellent

Quality of the paper: Is the overall quality of the paper suitable?

Excellent

Is the length of the paper justified?

Yes

Should the paper be seen by a specialist statistical reviewer?

No

Do you have any concerns about statistical analyses in this paper? If so, please specify them explicitly in your report.

No

It is a condition of publication that authors make their supporting data, code and materials available - either as supplementary material or hosted in an external repository. Please rate, if applicable, the supporting data on the following criteria.

Is it accessible?

N/A

Is it clear?

N/A

Is it adequate?

N/A

Do you have any ethical concerns with this paper?

No

Comments to the Author

This is an excellent review of some but not 'everything' that happens at the molecular and cellular levels after WGD (even if 'everything' changes at once!). Increased attention to WGD, especially in plants, has been evident in recent years, but generally the work focuses on the genome or transcriptome or ecological effects, so it is great to see a discussion of possible mechanisms of change at the cellular level. The awesome recent review by Doyle and Coate fills part of this void, but this paper very strategically selects some of the key processes that may operate to generate novelty, respond to stress, and more. I have relatively few comments for improvement.

Some of the responses described are related to changes in cell size; these nucleotypic changes can have cascading effects, but not all do, right? It would be good to note the importance of nucleotypic effects per se (see papers by Bennett, for example), which predated Levin's classic (1983) paper, which is cited.

How does tetraploidy - twice the genetic material - play into these scenarios? Is there more gene product that leads to these responses? and that may eventually be dialed down in established autopolyploids? What about increased heterozygosity of autopolyploids relative to diploids?

In contrast to allopolyploids, which have 2 distinct genomes that fractionate through time, autopolyploids don't have that mechanism for DNA loss and diploidization - so would we expect more change at the genetic level (selection, regulation) in autos than allos?

L 60. I don't think this necessarily has to occur in this order - unreduced gametes may be produced first so that WGD in a sense occurs first but the polyploid individual is formed with hybridization and WGD occurring simultaneously.

A few minor points are addressed below.

L 116. Insert of. number of polyploids

L 230. Remove comma after arenosa

L 233. space is needed before is

L 248. Slower cell cycles may relate to the fact that there is more DNA to replicate; this point has been reported previously.

L 375. This finding is odd, given the high incidence of polyploids at high latitudes and elevation.

L 380. Replace the period after further with a comma.

L 389. Change are to is or change subject to birch trees?

L 410. After (113) another) is needed.

L 484. WD should be WGD.

Other typos will likely be caught by the copy editor.

Review form: Reviewer 2

Recommendation

Accept with minor revision (please list in comments)

Scientific importance: Is the manuscript an original and important contribution to its field?

Good

General interest: Is the paper of sufficient general interest?

Good

Quality of the paper: Is the overall quality of the paper suitable?

Good

Is the length of the paper justified?

Yes

Should the paper be seen by a specialist statistical reviewer?

No

Do you have any concerns about statistical analyses in this paper? If so, please specify them explicitly in your report.

No

It is a condition of publication that authors make their supporting data, code and materials available - either as supplementary material or hosted in an external repository. Please rate, if applicable, the supporting data on the following criteria.

Is it accessible?

N/A

Is it clear?

N/A

Is it adequate?

N/A

Do you have any ethical concerns with this paper?

No

Comments to the Author

Review of 'When everything changes at once – Finding a new normal after genome duplication'

In the review paper entitled, 'When everything changes at once – Finding a new normal after genome duplication', the author discusses the effects of WGD on the biology of plants, and then mainly focusses on the effects on cell biology, such as the effects on cell size, cell types, and cell physiology. The author also discusses the effects of WGD on stress, and the short- and longer-term responses to WGD and how polyploids might evolve to a new conditionally adaptive, normal.

I have read this review with great interest. This review is, in my opinion, valuable because of the elegant and detailed integration of physiological and cellular and molecular responses associated with polyploidy. Although many review papers have been written in the recent past about WGDs and the effects thereof, this particular review paper is one of few that focusses on the cell-size effects of cells and cell types such as stomata, pollen tubes, and vasculature cells, and on cell growth and stress tolerance. In doing so, the author focusses on 'common' phenotypic changes associated with WGDs, by surveying tens of studies on different neo-autopolyploids and established paleo-autopolyploids. It is clear that there are several phenotypes common to WGD, and several are discussed in greater detail in the current review. Also the broad comparative survey is informative for shaping theories of the common features of adaptation after WGD, and, as the author mentions in the conclusion, to generate testable hypotheses at the molecular level. I have a few, I guess mainly minor, comments.

In discussing (common) traits that often change in response to WGD or polyploidy, the author puts forward the idea that types of responses to WGD fall into three different classes. In short: "type A" traits are those where a WGD-induced change is followed by an evolutionary response that returns the trait partly or wholly to a diploid-like value. Such traits would only be observable in neopolyploids. Examples include, for instance, leaf cell size, pollen tube growth rate, cell growth rate, and salt tolerance. "Type B" traits are those where WGD induces a change that is either maintained or further elaborated by evolution. Examples include pollen tube width, stomatal density, and drought tolerance. "Type C" traits would appear not as immediate consequences, but sometime after WGD. Where Type C traits appear repeatedly in independent polyploids, it is hypothesized that they might arise as parallel evolutionary responses to consistent immediate effects of WGD. These different types (A, B, and C) of responses are also visualized in Fig. 2. Personally, I think this is a great way to think of, or to classify, responses to WGD, but then found it a pity that, apart from introducing those classes early on in the manuscript, and in Fig. 2, further in the manuscript there is hardly any mentioning anymore of these concepts and classes. Only once more (except for the figure legend of Fig. 2), the author refers to a "Type A trait" (line 299). When I was reading the manuscript, and came to the part where these different types of traits or responses were introduced, I expected that, further in the review, I would see many more references to these concepts.

With respect to the detailed discussion on how WGDs affect traits such as cell size, in particular of guard cells, pollen tubes, and xylem, and stress resilience, I have no real comments, except than maybe that it would be nice to see which of those the author considers "Type A, B, or C traits/classes". Or are these all "Type A traits", as mentioned higher (basically in a single sentence (line 139))? Here, the author could make better use of the definitions introduced above, I would think, especially if the author hopes that these terminologies will find (or get adopted by) a broader audience.

Again, in the conclusion, the author writes: 'For example, it seems from this survey that drought tolerance is associated with both new and established tetraploids, while salt tolerance is almost exclusively reported in neotetraploids.' If I understood the author well, this would be THE place

to refer to the different Type traits again.

Since this entire review is about plants, maybe that should be specified in the title as well?

Line 46: 'Do the direct effects of WGD trigger evolutionary re-tuning?' It is not entirely clear to me what the author means by evolutionary re-tuning. Does the author mean adaptation to the direct effects of WGD or an adaptive response to the direct effects of WGD?

Lines 50-52. This paragraph is a little unclear. 'Some effects are likely mild'. It is unclear whether this concerns the effects of cell size changes, or the effects of WGD in this context.

Lines 123-124: 'decrease of DNA content over time, or other selected factors'. Maybe it is worthwhile to consider that decrease of DNA content over time could be both due to selection and neutral processes? Of course, little is known about these processes, but now the sentence implies that cell-size evolving downward is a consequence of selection (alone).

Lines 145-147: I object slightly to the interpretation of the conjectured 'Type C' traits. The author writes: 'we could hypothesize they might arise as parallel evolutionary responses to consistent immediate effects of WGD.' Why should those traits appear as a response to 'immediate effects' and could they not be a long-term consequence of WGD? For instance, subfunctionalization of duplicate genes could fall under the 'C' category, I guess, but this is not necessarily in response to immediate effects of WGD, it is simply a process that takes more time than e.g. cell-size changes.

Lines 189-191: 'Thus, we might expect that if neopolyploids, with their larger guard cells, have more sluggish responses, that this may be compensated for by evolution.' This is rather vague. In what way does evolution 'compensate' for sluggish response in larger guard cells? Perhaps it would be better to rephrase this as 'Larger guard cells in neopolyploids have more sluggish responses, which may be maladaptive and necessitate adaptation', or something like that (if I understood correctly).

Line 193: 'Interestingly, in the three genome analyses of polyploid adaptations available to date'. Might be worthwhile to add 'in plants' somewhere. Polyploid adaptations is also vague. This could be very broadly interpreted.

Lines 201, 206, 241, 274: 'modified by selection' and 'selection having targeted'. In what way? It might be nice to give some more detail here I think. In what way did the polyploids show signs of adaptation vs. the diploids, and how were these selective effects disentangled from demographic effects?. Elsewhere in the paragraph and next one, 'differentiation outliers' are mentioned, suggesting something based on F_{st} (?) is taken as evidence for selection, but of course genetic differentiation can be an effect of demography (and especially the likely bottlenecks associated with polyploid establishment). Are we speaking of 'differentiation outliers' throughout, or is there some special effort to infer natural selection in the cases where 'selection' is explicitly mentioned? It might be worthwhile to go in a little more detail or to explain the usage of these terms a little more.

Line 239: '[...] suggesting evolution can adjust this ratio to suit the habitat'. I think it is best to avoid talking about evolution as an 'active agent' that 'adapts organisms' to make them suit their place in nature. Maybe better write something like 'suggesting this ratio can be adjusted through an evolutionary response for local adaptation', or something similar.

Line 334: Should that doi link be here?

Lines 381-382: 'showing that evolution can further enhance conductivity through other means, building on the foundation set by WGD.' Another sentence where 'evolution' seems to be an acting agent.

Decision letter (RSPB-2020-2154.R0)

28-Sep-2020

Dear Kirsten

I am pleased to inform you that your Darwin Review RSPB-2020-2154 entitled "When everything changes at once – Finding a new normal after genome duplication" has been accepted for publication in Proceedings B. Not that we ever had any doubt...

The referees love it, as do I, but have also suggested some useful revisions to your manuscript. Therefore, I invite you to respond to the referees' comments and revise your manuscript. Because the schedule for publication is very tight, it is a condition of publication that you submit the revised version of your manuscript within 7 days. If you do not think you will be able to meet this date please let us know.

Best wishes,

Innes

Prof. Innes Cuthill
Reviews Editor, Proceedings B
mailto: proceedingsb@royalsociety.org

Reviewer(s)' Comments to Author:

Referee: 1

Comments to the Author(s)

This is an excellent review of some but not 'everything' that happens at the molecular and cellular levels after WGD (even if 'everything' changes at once!). Increased attention to WGD, especially in plants, has been evident in recent years, but generally the work focuses on the genome or transcriptome or ecological effects, so it is great to see a discussion of possible mechanisms of change at the cellular level. The awesome recent review by Doyle and Coate fills part of this void, but this paper very strategically selects some of the key processes that may operate to generate novelty, respond to stress, and more. I have relatively few comments for improvement.

Some of the responses described are related to changes in cell size; these nucleotypic changes can have cascading effects, but not all do, right? It would be good to note the importance of nucleotypic effects per se (see papers by Bennett, for example), which predated Levin's classic (1983) paper, which is cited.

How does tetraploidy - twice the genetic material - play into these scenarios? Is there more gene product that leads to these responses? and that may eventually be dialed down in established autopolyploids? What about increased heterozygosity of autopolyploids relative to diploids?

In contrast to allopolyploids, which have 2 distinct genomes that fractionate through time, autopolyploids don't have that mechanism for DNA loss and diploidization - so would we expect more change at the genetic level (selection, regulation) in autos than allos?

L 60. I don't think this necessarily has to occur in this order - unreduced gametes may be produced first so that WGD in a sense occurs first but the polyploid individual is formed with hybridization and WGD occurring simultaneously.

A few minor points are addressed below.

L 116. Insert of. number of polyploids

L 230. Remove comma after arenosa

L 233. space is needed before is

L 248. Slower cell cycles may relate to the fact that there is more DNA to replicate; this point has been reported previously.

L 375. This finding is odd, given the high incidence of polyploids at high latitudes and elevation.

L 380. Replace the period after further with a comma.

L 389. Change are to is or change subject to birch trees?

L 410. After (113) another) is needed.

L 484. WD should be WGD.

Other typos will likely be caught by the copy editor.

Referee: 2

Comments to the Author(s)

Review of 'When everything changes at once - Finding a new normal after genome duplication'

In the review paper entitled, 'When everything changes at once - Finding a new normal after genome duplication', the author discusses the effects of WGD on the biology of plants, and then mainly focusses on the effects on cell biology, such as the effects on cell size, cell types, and cell physiology. The author also discusses the effects of WGD on stress, and the short- and longer-term responses to WGD and how polyploids might evolve to a new conditionally adaptive, normal.

I have read this review with great interest. This review is, in my opinion, valuable because of the elegant and detailed integration of physiological and cellular and molecular responses associated with polyploidy. Although many review papers have been written in the recent past about

WGDs and the effects thereof, this particular review paper is one of few that focusses on the cell-size effects of cells and cell types such as stomata, pollen tubes, and vasculature cells, and on cell growth and stress tolerance. In doing so, the author focusses on ‘common’ phenotypic changes associated with WGDs, by surveying tens of studies on different neo-autopolyploids and established paleo-autopolyploids. It is clear that there are several phenotypes common to WGD, and several are discussed in greater detail in the current review. Also the broad comparative survey is informative for shaping theories of the common features of adaptation after WGD, and, as the author mentions in the conclusion, to generate testable hypotheses at the molecular level. I have a few, I guess mainly minor, comments.

In discussing (common) traits that often change in response to WGD or polyploidy, the author puts forward the idea that types of responses to WGD fall into three different classes. In short: “type A” traits are those where a WGD-induced change is followed by an evolutionary response that returns the trait partly or wholly to a diploid-like value. Such traits would only be observable in neopolyploids. Examples include, for instance, leaf cell size, pollen tube growth rate, cell growth rate, and salt tolerance. “Type B” traits are those where WGD induces a change that is either maintained or further elaborated by evolution. Examples include pollen tube width, stomatal density, and drought tolerance. “Type C” traits would appear not as immediate consequences, but sometime after WGD. Where Type C traits appear repeatedly in independent polyploids, it is hypothesized that they might arise as parallel evolutionary responses to consistent immediate effects of WGD. These different types (A, B, and C) of responses are also visualized in Fig. 2. Personally, I think this is a great way to think of, or to classify, responses to WGD, but then found it a pity that, apart from introducing those classes early on in the manuscript, and in Fig. 2, further in the manuscript there is hardly any mentioning anymore of these concepts and classes. Only once more (except for the figure legend of Fig. 2), the author refers to a “Type A trait” (line 299). When I was reading the manuscript, and came to the part where these different types of traits or responses were introduced, I expected that, further in the review, I would see many more references to these concepts.

With respect to the detailed discussion on how WGDs affect traits such as cell size, in particular of guard cells, pollen tubes, and xylem, and stress resilience, I have no real comments, except than maybe that it would be nice to see which of those the author considers “Type A, B, or C traits/classes”. Or are these all “Type A traits”, as mentioned higher (basically in a single sentence (line 139))? Here, the author could make better use of the definitions introduced above, I would think, especially if the author hopes that these terminologies will find (or get adopted by) a broader audience.

Again, in the conclusion, the author writes: ‘For example, it seems from this survey that drought tolerance is associated with both new and established tetraploids, while salt tolerance is almost exclusively reported in neotetraploids.’ If I understood the author well, this would be THE place to refer to the different Type traits again.

Since this entire review is about plants, maybe that should be specified in the title as well?

Line 46: ‘Do the direct effects of WGD trigger evolutionary re-tuning?’ It is not entirely clear to me what the author means by evolutionary re-tuning. Does the author mean adaptation to the direct effects of WGD or an adaptive response to the direct effects of WGD?

Lines 50-52. This paragraph is a little unclear. ‘Some effects are likely mild’. It is unclear whether this concerns the effects of cell size changes, or the effects of WGD in this context.

Lines 123-124: ‘decrease of DNA content over time, or other selected factors’. Maybe it is worthwhile to consider that decrease of DNA content over time could be both due to selection and neutral processes? Of course, little is known about these processes, but now the sentence implies that cell-size evolving downward is a consequence of selection (alone).

Lines 145-147: I object slightly to the interpretation of the conjectured 'Type C' traits. The author writes: 'we could hypothesize they might arise as parallel evolutionary responses to consistent immediate effects of WGD.' Why should those traits appear as a response to 'immediate effects' and could they not be a long-term consequence of WGD? For instance, subfunctionalization of duplicate genes could fall under the 'C' category, I guess, but this is not necessarily in response to immediate effects of WGD, it is simply a process that takes more time than e.g. cell-size changes.

Lines 189-191: 'Thus, we might expect that if neopolyploids, with their larger guard cells, have more sluggish responses, that this may be compensated for by evolution.' This is rather vague. In what way does evolution 'compensate' for sluggish response in larger guard cells? Perhaps it would be better to rephrase this as 'Larger guard cells in neopolyploids have more sluggish responses, which may be maladaptive and necessitate adaptation', or something like that (if I understood correctly).

Line 193: 'Interestingly, in the three genome analyses of polyploid adaptations available to date'. Might be worthwhile to add 'in plants' somewhere. Polyploid adaptations is also vague. This could be very broadly interpreted.

Lines 201, 206, 241, 274: 'modified by selection' and 'selection having targeted'. In what way? It might be nice to give some more detail here I think. In what way did the polyploids show signs of adaptation vs. the diploids, and how were these selective effects disentangled from demographic effects?. Elsewhere in the paragraph and next one, 'differentiation outliers' are mentioned, suggesting something based on F_{st} (?) is taken as evidence for selection, but of course genetic differentiation can be an effect of demography (and especially the likely bottlenecks associated with polyploid establishment). Are we speaking of 'differentiation outliers' throughout, or is there some special effort to infer natural selection in the cases where 'selection' is explicitly mentioned? It might be worthwhile to go in a little more detail or to explain the usage of these terms a little more.

Line 239: '[...] suggesting evolution can adjust this ratio to suit the habitat'. I think it is best to avoid talking about evolution as an 'active agent' that 'adapts organisms' to make them suit their place in nature. Maybe better write something like 'suggesting this ratio can be adjusted through an evolutionary response for local adaptation', or something similar.

Line 334: Should that doi link be here?

Lines 381-382: 'showing that evolution can further enhance conductivity through other means, building on the foundation set by WGD.' Another sentence where 'evolution' seems to be an acting agent.

Author's Response to Decision Letter for (RSPB-2020-2154.R0)

See Appendix A.

Decision letter (RSPB-2020-2154.R1)

26-Oct-2020

Dear Dr Bomblies

I am pleased to inform you that your manuscript entitled "When everything changes at once – Finding a new normal after genome duplication" has been accepted for publication in Proceedings B.

If you are likely to be away from e-mail contact during this period, let us know. Due to rapid publication and an extremely tight schedule, if comments are not received, we may publish the paper as it stands.

Your article has been estimated as being 15 pages long. Our Production Office will be able to confirm the exact length at proof stage.

Open access

You are invited to opt for open access via our author pays publishing model. Payment of open access fees will enable your article to be made freely available via the Royal Society website as soon as it is ready for publication. For more information about open access publishing please visit our website at http://royalsocietypublishing.org/site/authors/open_access.xhtml.

The open access fee is £1,700 per article (plus VAT for authors within the EU). If you wish to opt for open access then please let us know as soon as possible.

Paper charges

Sincerely,
Proceedings B
<mailto:proceedingsb@royalsociety.org>

Appendix A

Reviewer(s)' Comments to Author:

Referee: 1

Comments to the Author(s)

This is an excellent review of some but not 'everything' that happens at the molecular and cellular levels after WGD (even if 'everything' changes at once!). Increased attention to WGD, especially in plants, has been evident in recent years, but generally the work focuses on the genome or transcriptome or ecological effects, so it is great to see a discussion of possible mechanisms of change at the cellular level. The awesome recent review by Doyle and Coate fills part of this void, but this paper very strategically selects some of the key processes that may operate to generate novelty, respond to stress, and more. I have relatively few comments for improvement. *Thank you!*

Some of the responses described are related to changes in cell size; these nucleotypic changes can have cascading effects, but not all do, right? It would be good to note the importance of nucleotypic effects per se (see papers by Bennett, for example), which predated Levin's classic (1983) paper, which is cited.

How does tetraploidy - twice the genetic material - play into these scenarios? Is there more gene product that leads to these responses? and that may eventually be dialed down in established autopolyploids? What about increased heterozygosity of autopolyploids relative to diploids?

These are good points, but outside the scope here. I think that having twice the genetic material contributes (somehow) to the increase in cell size, which is the trait of interest here. But the connection is still hotly debated. I don't think that the additional gene product leads to the responses observed here (if anything, I would think it helps make them milder, by scaling protein content to cell size to some extent).

More gene product: seems so in the studies that have looked at it – it seems to scale with DNA content in most cases and I do not really have a good theory for how it could really play much role in the traits discussed here, except that it may contribute to cell size or help mitigate some of the scaling effects, but this has not been investigated, to my knowledge.

Heterozygosity: I think the increased heterozygosity may be relevant only in providing additional substrate for evolution to act on. I think cell size effects are independent of heterozygosity.

In contrast to allopolyploids, which have 2 distinct genomes that fractionate through time, autopolyploids don't have that mechanism for DNA loss and diploidization - so would we expect more change at the genetic level (selection, regulation) in autos than allos?

Perhaps, yes – it's certainly a good idea, but this has not, to my knowledge been directly tested.

L 60. I don't think this necessarily has to occur in this order - unreduced gametes may be produced first so that WGD in a sense occurs first but the polyploid individual is formed with hybridization and WGD occurring simultaneously.

I modified this to read "...while allopolyploids can form multiple ways, e.g. from hybridization events followed by WGD, or from fusion of unreduced gametes from genetically distinct parents."

A few minor points are addressed below.

L 116. Insert of. number of polyploids

I refer instead to the tables that give these numbers.

L 230. Remove comma after arenosa

Done

L 233. space is needed before is

Done

L 248. Slower cell cycles may relate to the fact that there is more DNA to replicate; this point has been reported previously.

Actually, while in diploids DNA content scales with DNA content, it seems that diploids and tetraploids do not commonly differ in mitotic cell cycle duration. I have added a sentence to note this.

L 375. This finding is odd, given the high incidence of polyploids at high latitudes and elevation.

It is odd, I don't have a good explanation for it...

L 380. Replace the period after further with a comma.

Done

L 389. Change are to is or change subject to birch trees?

Done

L 410. After (113) another) is needed.

Done

L 484. WD should be WGD.

Done

Other typos will likely be caught by the copy editor.

Several others also fixed.

Referee: 2

Comments to the Author(s)

Review of 'When everything changes at once – Finding a new normal after genome duplication'

In the review paper entitled, 'When everything changes at once – Finding a new normal after genome duplication', the author discusses the effects of WGD on the biology of plants, and then mainly focusses on the effects on cell biology, such as the effects on cell size, cell types, and cell physiology. The author also discusses the effects of WGD on stress, and the short- and longer-term responses to WGD and how polyploids might evolve to a new conditionally adaptive, normal.

I have read this review with great interest. This review is, in my opinion, valuable because of the elegant and detailed integration of physiological and cellular and molecular responses associated with polyploidy. Although many review papers have been written in the recent past about WGDs and the effects thereof, this particular review paper is one of few that focusses on the cell-size effects of cells and cell types such as stomata, pollen tubes, and vasculature cells, and on cell growth and stress tolerance. In doing so, the author focusses on 'common' phenotypic changes associated with WGDs, by surveying tens of studies on different neo-autopolyploids and established paleo-autopolyploids. It is clear that there are several phenotypes common to WGD, and several are discussed in greater detail in the current review. Also the broad comparative survey is informative for shaping theories of the common features of adaptation after WGD, and, as the author mentions in the conclusion, to generate testable hypotheses at the molecular level. I have a few, I guess mainly minor, comments.

Thank you!

In discussing (common) traits that often change in response to WGD or polyploidy, the author puts forward the idea that types of responses to WGD fall into three different classes. In short: "type A" traits are those where a WGD-induced change is followed by an evolutionary response that returns the trait partly or wholly to a diploid-like value. Such traits would only be observable in neopolyploids. Examples include, for instance, leaf cell size, pollen tube growth rate, cell growth rate, and salt tolerance. "Type B" traits are those where WGD induces a change that is either maintained or further elaborated by evolution. Examples include pollen tube width, stomatal density, and drought tolerance. "Type C" traits would appear not as immediate consequences, but sometime after WGD. Where Type C traits appear repeatedly in independent polyploids, it is hypothesized that they might arise as parallel evolutionary responses to consistent immediate effects of WGD. These different types (A, B, and C) of responses are also visualized in Fig. 2. Personally, I think this is a great way to think of, or to classify, responses to WGD, but then found it a pity that, apart from introducing those classes early on in the manuscript, and in Fig. 2, further in the manuscript there is hardly any mentioning anymore of these concepts and classes. Only once more (except for the figure legend of Fig. 2), the author refers to a "Type A trait" (line 299). When I was reading the manuscript, and came to the part where these different types of traits or responses were introduced, I expected that, further in the review, I would see many more references to these concepts.

With respect to the detailed discussion on how WGDs affect traits such as cell size, in particular of guard cells, pollen tubes, and xylem, and stress resilience, I have no real comments, except than maybe that it would be nice to see which of those the author considers "Type A, B, or C traits/classes". Or are these all "Type A traits", as mentioned higher (basically in a single sentence (line 139))? Here, the author could make better use of the definitions introduced above, I would think, especially if the author hopes that these terminologies will find (or get adopted by) a broader audience.

Again, in the conclusion, the author writes: 'For example, it seems from this survey that drought tolerance is associated with both new and established tetraploids, while salt tolerance is almost exclusively reported in neotetraploids.' If I understood the author well, this would be THE place to refer to the different Type traits again.

Yes, thank you – I have added this, as well as reference to trait types throughout.

Since this entire review is about plants, maybe that should be specified in the title as well?

I would like to respectfully decline to change the title – it is catchier this way, and more importantly, the things I talk about (broad concepts) will be relevant to other things as well - often papers with "plant" in the title get read far less.

Line 46: 'Do the direct effects of WGD trigger evolutionary re-tuning?' It is not entirely clear to me what the author means by evolutionary re-tuning. Does the author mean adaptation to the direct effects of WGD or an adaptive response to the direct effects of WGD?

Reworded to: "Do the effects of WGD trigger a longer-term re-tuning of the physiology of the organism?"

Lines 50-52: This paragraph is a little unclear. 'Some effects are likely mild'. It is unclear whether this concerns the effects of cell size changes, or the effects of WGD in this context.

Added "of cell size change"

Lines 123-124: 'decrease of DNA content over time, or other selected factors'. Maybe it is worthwhile to consider that decrease of DNA content over time could be both due to selection and neutral processes? Of course, little is known about these processes, but now the sentence implies that cell-size evolving downward is a consequence of selection (alone).

I removed "selected" and added the phrase, "which could be adaptive or selectively neutral"

Lines 145-147: I object slightly to the interpretation of the conjectured 'Type C' traits. The author writes: 'we could hypothesize they might arise as parallel evolutionary responses to consistent immediate effects of WGD.' Why should those traits appear as a response to 'immediate effects' and could they not be a long-term consequence of WGD? For instance, subfunctionalization of duplicate genes could fall under the 'C' category, I guess, but this is not necessarily in response to immediate effects of WGD, it is simply a process that takes more time than e.g. cell-size changes.

I removed the implication of immediate effects.

Lines 189-191: 'Thus, we might expect that if neopolyploids, with their larger guard cells, have more sluggish responses, that this may be compensated for by evolution.' This is rather vague. In what way does evolution 'compensate' for sluggish response in larger guard cells? Perhaps it would be better to rephrase this as 'Larger guard cells in neopolyploids have more sluggish responses, which may be maladaptive and necessitate adaptation', or something like that (if I understood correctly).

I added several clarifying sentences

Line 193: 'Interestingly, in the three genome analyses of polyploid adaptations available to date'. Might be worthwhile to add 'in plants' somewhere. Polyploid adaptations is also vague. This could be very broadly interpreted.

I reworded this phrase to "...analyses of adaptation to polyploidy available to date in plants..."

Lines 201, 206, 241, 274: 'modified by selection' and 'selection having targeted'. In what way? It might be nice to give some more detail here I think. In what way did the polyploids show signs of adaptation vs. the diploids, and how were these selective effects disentangled from demographic effects?. Elsewhere in the paragraph and next one, 'differentiation outliers' are mentioned, suggesting something based on F_{st} (?) is taken as evidence for selection, but of course genetic differentiation can be an effect of demography (and especially the likely bottlenecks associated with polyploid establishment). Are we speaking of 'differentiation outliers' throughout, or is there some special effort to infer natural selection in the cases where 'selection' is explicitly mentioned? It might be worthwhile to go in a little more detail or to explain the usage of these terms a little more.

I added a brief description of the studies from the three species to the introduction, and where I previously said "selection" have altered the wording to clarify.

Line 239: [...] suggesting evolution can adjust this ratio to suit the habitat'. I think it is best to avoid talking about evolution as an 'active agent' that 'adapts organisms' to make them suit their place in nature. Maybe better write something like 'suggesting this ratio can be adjusted through an evolutionary response for local adaptation', or something similar.

Good point – I have reworded all implications of evolution as an active agent – it was just sloppy wording and not meant to read that way.

Line 334: Should that doi link be here?

No, sorry, it was meant to be a reference and has been changed.

Lines 381-382: 'showing that evolution can further enhance conductivity through other means, building on the foundation set by WGD.' Another sentence where 'evolution' seems to be an acting agent.

Changed.